# Quality of Life in Myasthenia Gravis and Correlation of MG-QOL15 with Other Functional Scales

**DOI:** 10.3390/jcm11082189

**Published:** 2022-04-14

**Authors:** Laura Diez Porras, Christian Homedes, Maria Antonia Alberti, Valentina Velez Santamaria, Carlos Casasnovas

**Affiliations:** 1Neurometabolic Diseases Group, Bellvitge Biomedical Research Institute (IDIBELL), 199 Granvia de l’Hospitalet, 08908 L’Hospitalet de Llobregat, Spain; diez.laura.3@gmail.com (L.D.P.); christianhomedes@gmail.com (C.H.); aalberti@bellvitgehospital.cat (M.A.A.); pvelezsantamaria@bellvitgehospital.cat (V.V.S.); 2Neuromuscular Unit, Department of Neurology, Bellvitge University Hospital, Feixa Llarga Street n/n, 08907 L’Hospitalet del Llobregat, Spain; 3Center for Biomedical Research on Rare Diseases (CIBERER), ISCIII, 3–5 Monforte de Lemos, Pabellón 121, 28029 Madrid, Spain

**Keywords:** quality of life, myasthenia gravis, intravenous immunoglobulins, prednisone, scales, questionnaires, MG-QOL15

## Abstract

Health-related quality of life (HRQOL) in myasthenia gravis (MG) is frequently decreased. Further, there are many validated clinical scales and questionnaires to evaluate the clinical status in MG. We aimed to determine if there was an improvement in HRQOL following an intensive treatment for MG, identify which demographic and clinical features influenced patients’ HRQOL, and investigate if the questionnaire MG-QOL15 correlated with other evaluation scales. We recruited 45 patients with generalised MG who were starting immunomodulatory treatment with intravenous immunoglobulins and prednisone for the first time. At each visit, we administered several validated scales for MG. The mean MG-QOL15 score improved significantly at 4 and 6 weeks of the study. Additionally, the MG-QOL15 score correlated strong with the Myasthenia Gravis-Activities of Daily Living (MG-ADL) and the Neuro-QOL Fatigue and weakest with the Quantitative Myasthenia Gravis Scoring System (QMG). The QMG score prior to study enrolment was associated with HRQOL. We observed that HRQOL in MG improved after receiving an intensive immunomodulatory treatment and achieving better control of the symptoms. The questionnaire MG-QOL15 correlated positively with other clinical measures. As MG is a fluctuating condition, and some symptoms are difficult to examine, we direct physicians toward the use of scales and questionnaires composed of items perceived by the patient.

## 1. Introduction

Myasthenia gravis (MG) is a chronic autoimmune disease in which the postsynaptic membrane of the neuromuscular junction is altered. Clinically, it produces fatigability and weakness of striated muscle. Characteristically, symptoms fluctuate throughout the day and improve after muscle rest. The clinical course of the disease involves exacerbations and remissions [1].

Currently, the aims of MG treatment are (1) to improve symptoms using acetylcholinesterase inhibitors, immunomodulators, plasmapheresis, and/or thymectomy; (2) to avoid or minimise the side effects of long-term medication; and (3) to restore the patient’s health-related quality of life (HRQOL) to its previous level. Recently, combined treatment with intravenous immunoglobulins (IVIg) and high-dose prednisone was found to be safe and effective in controlling symptoms of the disease [2]. In general, with appropriate treatment, most patients stabilise and are fully able to carry out their activities of daily living (ADLs). However, several studies have shown that patients with MG have a reduced HRQOL, as the disease can affect vision, speech, swallowing, ADLs, and physical tasks [1,3].

Some instruments are available for the evaluation of HRQOL in MG, such as the questionnaire MG-QOL15, which requires little time to administer and is easy to interpret [4]. This questionnaire comprises 15 questions on how disease symptoms affect the patient’s mood, ADLs, work, and social activities (see Appendix A). The questionnaire has been demonstrated to have good reliability and longitudinal validity [5] and has been used in several studies of HRQOL in MG [6,7,8,9]. There are also many scales available for the assessment of the patient with MG, including the Myasthenia Gravis-Activities of Daily Living Profile (MG-ADL), the Myasthenia Gravis-Composite Scale (MGC), the Quantitative Myasthenia Gravis Scoring System (QMG), and the Neuro-QOL Fatigue subscale. The MG-ADL is an eight-item questionnaire focusing on symptoms that are relevant to the patient with MG [10,11]. The MGC is an assessment tool that combines subjective items as perceived by the patient with MG and objective findings from clinical examination [12]. The QMG, in turn, is a scale to evaluate the patient’s clinical status using quantitative tests and spirometry [13]. The Neuro-QOL Fatigue subscale is a sub-section of the Neuro-QoL that focuses on fatigue and is completed by the patient [14]. Thus, there are multiple scales available, and in routine medical visits there is not generally enough time to complete them all, so it would be ideal to use fewer scales and determine how they correlate with each other. As fatigability and the fluctuation of symptoms are common in MG, in recent decades, scales involving patient-perceived items have gained importance. Some studies have already shown a good correlation for total scores among these specific MG scales and with MG-QOL15 [4,10,12,15] and a good correlation with score changes from the initial assessment [15,16].

The aims of this study were, firstly, to determine if there was a significant improvement in HRQOL following an intensive treatment for MG and identify which demographic and clinical features from our sample influenced patients’ HRQOL and, secondly, to determine if the MG-QOL15 score correlated with other clinical evaluation scales (MG-ADL, QMG, MGC, and Neuro-QOL Fatigue subscale). 

## 2. Material and Methods

### 2.1. Study Design

We designed a post-authorization, experimental, single-centre study with prospective follow-up. In line with the Declaration of Helsinki, the study protocol was approved by the institutional review board of the Clinical Research and Clinical Trials Unit (UCICEC, by its initials in Spanish) of the Bellvitge Biomedical Research Institute (IDIBELL). All participants signed an informed consent form, and the anonymity of the participants was preserved with the use of codes that were stored in a locked area. The UCICEC IDIBELL carried out regular monitoring of the study. 

### 2.2. Patients

Between April 2016 and January 2019, we recruited all consecutive patients who attended our hospital and who were aged older than 18 years with a diagnosis of generalised MG, in classes IIA to V of the Myasthenia Gravis Foundation of America (MGFA) clinical classification system. We included patients who were starting immunomodulatory treatment for MG for the first time who did not have other medical conditions that would interfere with treatment with prednisone or IVIg or that the investigator considered important. Pregnant or breastfeeding individuals were excluded (see inclusion and exclusion criteria at Appendix A). Patients received one round of IVIg (0.4 g/kg/day for 5 days) and, at 7–10 days, they started high-dose prednisone (1 mg/kg/day or 0.75 mg/kg/day if they had comorbidities). During the study, there was close medical follow-up with in-person visits just before starting prednisone and at 4 weeks and 6 weeks after starting prednisone, which included clinical examination, blood tests, and a battery of clinically validated evaluation scales and questionnaires (QMG, MGC, MG-ADL, and MG-QOL15). In the weeks with no scheduled in-person visits, there was a telephone visit in which patients were able to ask questions and the MG-ADL scale was administered.

### 2.3. Statistical Analysis

Data analysis was performed using the statistical package R version 4.0.3 (Auckland, New Zeland). We analysed changes in HRQOL from the basal visit to the visits at 4 and 6 weeks using a mixed linear model of the effect of the visit on HRQOL, taking into account the random effect of the patient due to repeated measures. To compare the HRQOL at each visit, a paired post hoc test was used. The Pearson correlation coefficient was used to study the correlation between HRQOL and other clinical measures (MG-ADL, MGC, QMG, and Neuro-QOL Fatigue) at the first visit and at 4 and 6 weeks. We analysed if there was a correlation between HRQOL at the first visit and at 4 weeks according to the following categorical variables: MGFA classification, bulbar involvement, presence or absence of anti-RACh antibodies, and presence or absence of anti-striated muscle antibodies. We also analysed the association between demographic factors and pre-existing symptoms and quality of life at the first visit. We calculated linear models of the effect of age, sex, previous QMG, and thymectomy prior to study enrolment. The conditions of use of the models were validated, and the 95% confidence intervals of the estimator were calculated whenever possible.

## 3. Results

### 3.1. Sample Description

A total of 47 patients were enrolled. Two patients left the study: one because she deteriorated following chemotherapy for the treatment of thymoma and a second because she had progressive bulbar palsy rather than MG. More than two thirds of our patients had their first presentation of MG, and most of the sample were men older than 65 years with anti-RACh antibodies (see Table 1).

### 3.2. Evolution of HRQOL throughout the Study

We observed, firstly, that the patients’ HRQOL improved significantly at 6 weeks of starting intensive treatment with IVIg and high-dose prednisone (Table 2). We also observed, on post hoc analysis, that patients’ HRQOL continued to improve significantly between 4 and 6 weeks (Figure 1). 

At the baseline visit, the mean MG-QOL15 score was 25.93. At 4 and 6 weeks of starting intensive treatment, the MG-QOL15 score was 14.91 and 10.53, respectively (Table 2). At 6 weeks, almost half of the patients had an MG-QOL15 score lower than 10 (Figure 2).

At visit one, before starting prednisone, the HRQOL scores measured using the MG-QOL15 were very disperse, the interquartile range was wide, the mean was over 20, and several patients scored higher than 40. In contrast, at the visits at 4 and 6 weeks from starting prednisone, the interquartile range of the MG-QOL15 score narrowed, the mean score decreased to less than 20, and several patients had a score less than 10.

HRQOL questionnaire MG-QOL15 gives a score that ranges from 0 to 60, with a higher score indicating a worse HRQOL. On the left of the figure is the histogram of HRQOL density at the first visit as measured with MG-QOL15: scores were heterogeneous, and the peak of the density curve was around 20. On the right, at 6 weeks, after having received intensive treatment for MG, most of the patients scored lower than 20, and the peak of the density curve was below 10.

### 3.3. Correlation of the MG-QOL15 with Other Functional Scales

We also found that the MG-QOL15 score correlated significantly with the scores on the clinical evaluation scales MG-ADL, QMG, MGC, and the Neuro-QOL Fatigue subscale at all visits. The strongest correlation was with the MG-ADL scale and the Neuro-QOL Fatigue, and the weakest correlation was with the QMG scale (Figure 3).

Scatter plots for the correlation of the questionnaire MG-QOL15 with the scales MG-ADL, QMG, MGC, and Neuro-QOL Fatigue are presented below. At all visits, the correlation between MG-QOL15 and the rest of the scales is positive; the trend line is steeper, and thus the association is stronger between MG-QOL and ADL at 4 and 6 weeks and the Neuro-QOL Fatigue subscale at 4 weeks. 

### 3.4. Factors Related to HRQOL

Of the different demographic and clinical factors studied, we found that QMG score prior to study enrolment, before receiving IVIg and prednisone, was significantly associated with HRQOL in our patients (*p* = 0.047). However, the remaining demographic and clinical factors studied did not affect HRQOL (Figure 4; age, *p* = 0.839; sex, *p* = 0.986; thymectomy, *p* = 0.163; presence of anti-RACh antibody, *p* = 0.116; presence of anti-striated muscle antibody, *p* = 0.765; MGFA classification, *p* = 0.418; and bulbar involvement, *p* = 0.599).

We carried out ANOVA and found no differences in HRQOL measured using MG-QOL15 and MGFA classification and the presence or absence of anti-striated muscle antibody.

## 4. Discussion

In MG, as in other chronic diseases, patients’ HRQOL is reduced. Quality of life improves with better control of the disease [9,17] and worsens with more severe disease [6,17,18,19]. Other factors that reduce HRQOL are bulbar involvement [8,17,20], generalised disease, and refractory disease [7]. In our study, we also found that disease severity according to the QMG scale prior to enrolment in the study was associated with HRQOL; however, MGFA classification did not affect the HRQOL of our patients, as this classification reflects the moment of greatest severity in the course of the disease but not the severity during the period of study [18,21].

Previous studies have found no association between HRQOL and the type of MG according to the age of onset, type of autoantibodies, or thymus histology [8,22], although it has been observed that the presence of thymoma [6], female sex [8], and older age at the time of assessment can have a negative effect [4,6,23,24]. We did not find differences in HRQOL based on the age, sex, or type of antibody in the patients studied. The presence of anti-RYR antibody has previously been noted to be indicative of more severe disease and the presence of thymoma [25]. Almost half of our patients had anti-striated muscle antibodies, and we thought that its presence would be associated with a worse HRQOL. However, we found that these antibodies did not affect HRQOL, possibly due to the low number of patients included and the fact that most of the sample pertained to the subgroup of MG with “very late” onset, in which, although the onset of the disease may be severe, it usually has a good response to treatment [26]. Other factors that have been found to affect HRQOL in MG are anxiety and depression [6,9], social support, occupation, educational level, and marital status [6,27,28].

With regard to the effect of immunomodulatory treatment in HRQOL, a study found that treatment with IVIg or plasmapheresis improved the HRQOL of MG patients at 14 days [29]. Concerning the association between corticosteroids and HRQOL in MG, previous studies have shown disparate results. Some studies have found that the daily dose of prednisone [17,18] and the total dose of prednisone [17] worsen HRQOL. In contrast, other studies found no association between the current use of prednisolone or the average prednisolone dose in the previous 3 months and the MG-QOL15 and proposed that this association may have a J-shape curve [9]: at the beginning of treatment with corticosteroids, HRQOL improves due to the improvement in myasthenia, but after taking them for a prolonged period, HRQOL worsens due to the cumulative side effects [9]. However, so far, there was no evidence that intensive therapy with IVIg and full-dose prednisone significantly improved HRQOL in MG, and we found an increase in HRQOL in MG using this therapeutic strategy. In fact, it has already been shown that an effective therapeutic intervention significantly improves patients’ HRQOL. One prospective study showed that those patients who improved within the MGFA-PIS classification (met the favourable or minimal manifestations criteria with doses of prednisolone < 5 mg/day) also had an increase in HRQOL, and those who worsened (no longer in pharmacologic remission) had a significant decrease in perceived HRQOL [17]. In addition, in the clinical trial REGAIN, Eculizumab improved the quality of life from the fourth week [30], and this positive effect was maintained in the extension phase of the trial [31]. Therefore, we consider that the substantial enhancement of HRQOL of our patients was due to the clinical improvement following the intensive treatment with IVIg and prednisone.

Several studies have shown a good correlation between various clinical measures of MG and HRQOL [4,5,7,22,32]. Burns et al. found a good correlation between MG-QOL15 score and MG-ADL, MGC, and MG-specific Manual Motor Test [4]. Later, in a study to validate the psychometric properties of the MG-QOL15 questionnaire, they found a strong correlation with MG-ADL (0.76) and MGC (0.67) and one slightly less so with the MG-specific Manual Motor Test (0.54) [5]. Longitudinally, a change in MG-QOL15 score has also been associated with changes in MGC score (0.53) [15]. Another prospective study using online surveys of 773 patients also found that MG-ADL and MG-QOL15 were strongly correlated (r = 0.77) [7]. The REGAIN clinical trial also found a strong correlation between change in the Neuro-QOL Fatigue subscale and change in MG-QOL15, which was more pronounced in patients in the treatment group; that is, fatigue was associated with a worse HRQOL and this improved in both groups of patients, particularly the group of patients that received treatment [33]. In our study, HRQOL measured using the MG-QOL15 also strongly correlated with the MG-ADL and the Neuro-QOL Fatigue subscale and had a moderate correlation with the MGC and slight correlation with the QMG physical examination scale. This result was to be expected and is consistent with previous studies, as the MG-QOL15, as well as the MG-ADL and Neuro-QOL Fatigue subscale, is composed exclusively of patient-perceived subjective items based on disease symptoms. In contrast, the QMG is a long scale, composed exclusively of objective clinical examination findings at a specific point in time and requires specific instruments (e.g., dynamometer and spirometer). Generally, in routine medical visits there is little time for examination, and not all health centres have the specific materials needed to perform the QMG. Furthermore, some MG symptoms are difficult to examine (e.g., dysphagia and chewing fatigability); symptoms fluctuate throughout the day, and it is difficult to assess the overall status of the disease at a specific point. Assessing the patient’s impression in MG is important, and the correlation between the different scales supports the use of scales that include subjective items and scales that are faster to use, and easy to interpret (such as MG-QOL15, MG-ADL, or MGC). 

Our study has limitations, such as the lack of a control group without intensive treatment, the small sample size, and the fact that we did not assess the presence of comorbidities such as anxiety and depression. Additionally, corticosteroids can produce a euphoric effect or a wellbeing effect in some patients, which probably influenced some patients to score higher for certain items on the MG-QOL15 questionnaire. Further, there is not a validated version of the MG-QOL15 translated to Spanish, and most of our patients do not have a good-enough level of English that would allow them to complete the questionnaire by themselves, so we used the English version of the MG-QOL15, and each item was translated verbally to Spanish, which could have caused interpretation errors. However, several studies have also used this questionnaire read aloud or translated verbally without having a validated translation available [5,19,33]. In addition, it would have been preferable to use the revised version, MG-QOL15r. However, the study predates the publication of the MG-QOL15r [34].

Myasthenia gravis can have a negative effect on patients’ physical, psychological, and social wellbeing. As symptoms can fluctuate on a daily basis or over different periods of time, it would be interesting to conduct a longer prospective study to evaluate changes in the HRQOL of patients with MG over time. Several experts have emphasised the importance of measuring HRQOL; however, the high pressure of time in some health centres makes routine assessment difficult. We think that to ensure a holistic health model, HRQOL should be assessed in addition to other clinical examinations, which would help guide decision-making, provide additional measures of the impact of treatment, and improve the doctor–patient relationship. It would also be interesting to determine more precisely which factors are associated with HRQOL in MG, as several predictors of poor HRQOL are modifiable (such as disease severity, anxiety, depression, social support, or occupation) and, in doing so, we could implement measures to improve HRQOL in addition to conventional medical treatment.

In conclusion, HRQOL in patients with generalised MG receiving immunomodulatory treatment for the first time for symptom control improved significantly at 4 and 6 weeks after receiving intensive treatment with IVIg and high-dose prednisone. The questionnaire MG-QOL15 had a positive correlation with other measures of clinical assessment (MG-ADL, MGC, QMG, and Neuro-QOL Fatigue). 

## Figures and Tables

**Figure 1 jcm-11-02189-f001:**
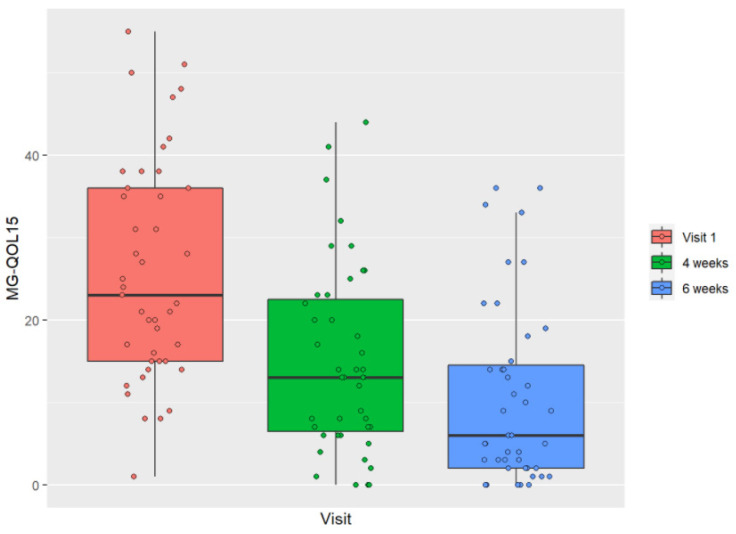
Graphical evolution of HRQL over the course of the study with boxplots.

**Figure 2 jcm-11-02189-f002:**
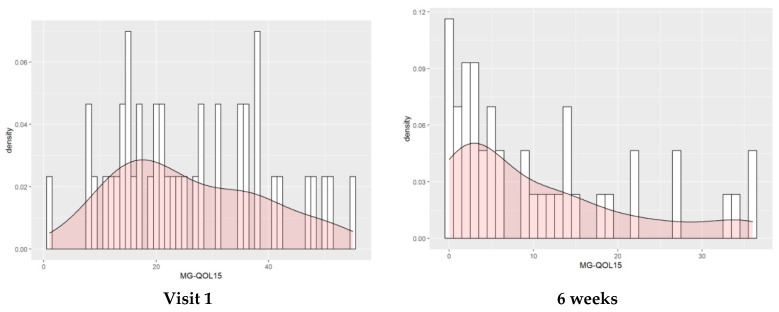
Histogram and density of the HRQOL distribution in our sample at visit 1 and at 6 weeks.

**Figure 3 jcm-11-02189-f003:**
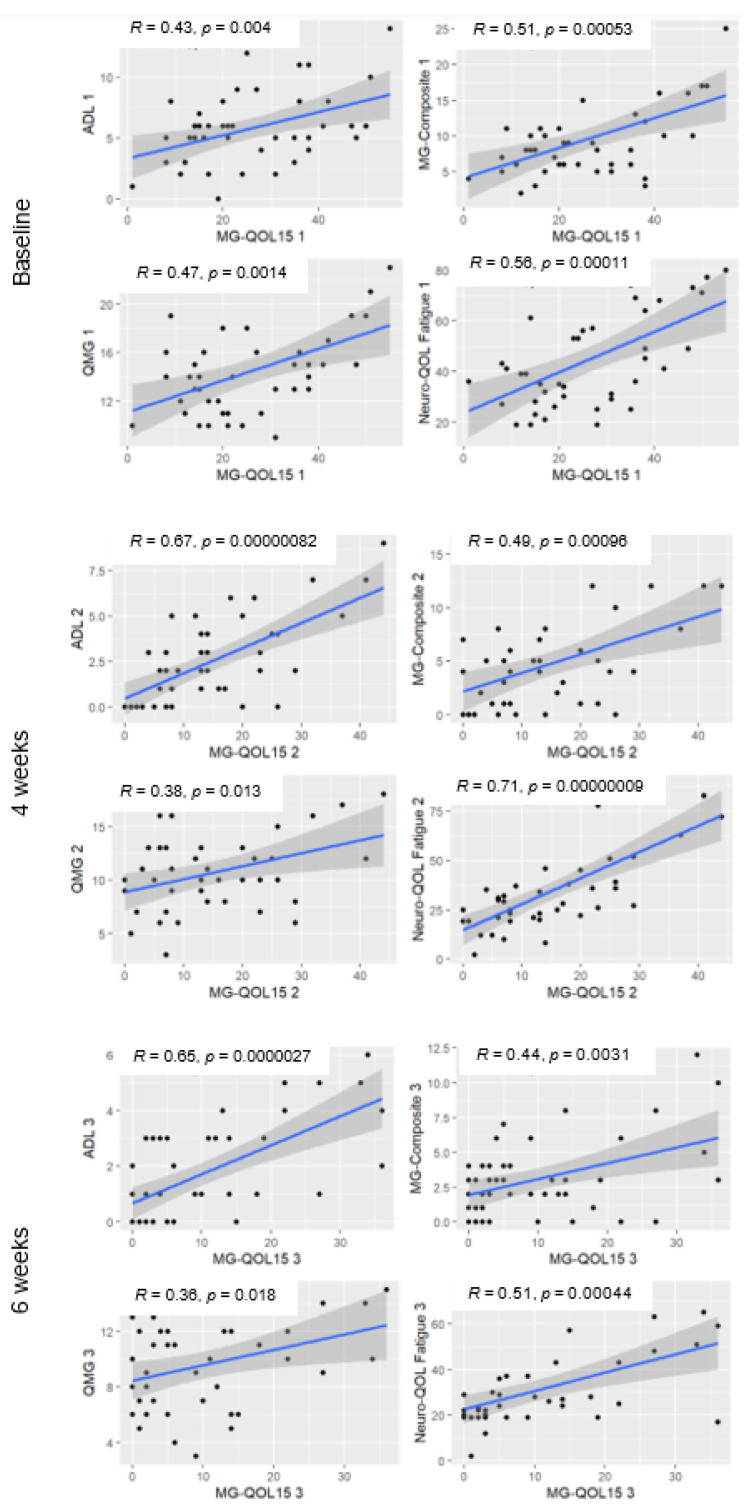
Correlations between the different clinical scales and the MG-QOL15 at visits 1, 5 (4 weeks), and 7 (6 weeks).

**Figure 4 jcm-11-02189-f004:**
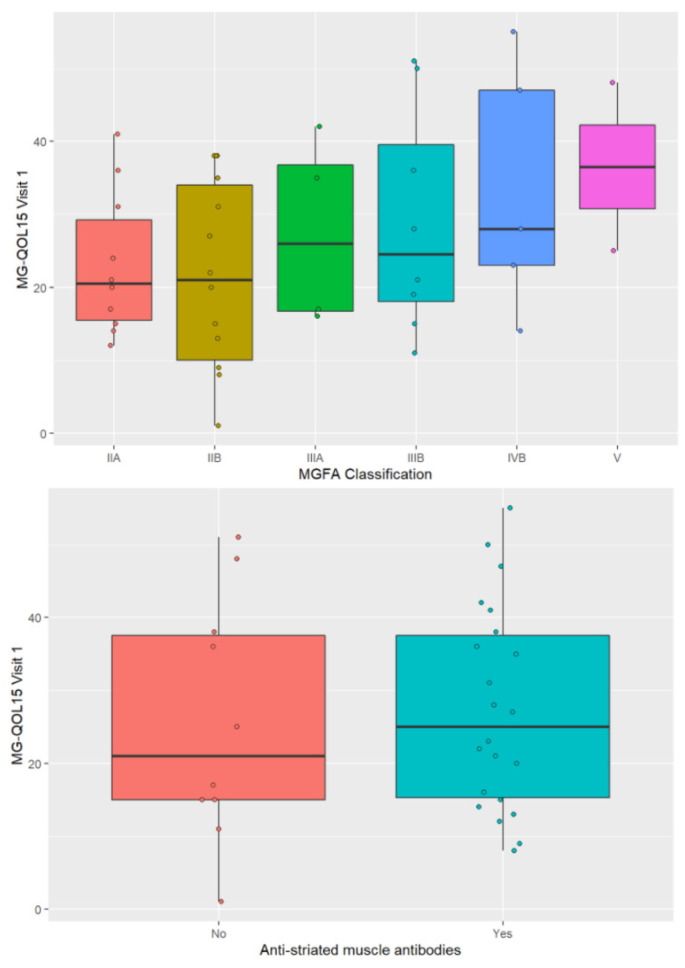
Representation of HRQOL according to the MGFA classification (worst MG status) and according to the presence of anti-striated muscle antibodies.

**Table 1 jcm-11-02189-t001:** Demographic characteristics of our population.

		Total (N)
Sex		45
Woman, *n* (%)	15 (33.33)	
Man, *n* (%)	30 (66.67)	
Age		
Minimum	26	
Maximum	85	
Median	69	
Age at onset, mean (standard deviation)	62.22 (16.32)	
MGFA class		
I, *n* (%)	0 (0)	
IIA, *n* (%)	10 (22.2)	
IIB, *n* (%)	14 (31.1)	
IIIA, *n* (%)	5 (11.1)	
IIIB, *n* (%)	9 (20)	
IV, *n* (%)	5 (11.1)	
V, *n* (%)	2 (4.4)	
Antibodies		
Anti-RAch, *n* (%)	41 (91.1)	
	0 (0)	
Anti-MuSK, *n* (%)	4 (8.9)	
Double seronegative, *n* (%)	22 (48.9)	
Anti-striated muscle, *n* (%)		
Thymus		
Thymoma, *n* (%)	2 (4.4)	
Thymic hyperplasia, *n* (%)	4 (8.9)	
Atrophy or CT without evidence of thymoma, *n* (%)	39 (86.7)	
Score on scales (pretreatment and at V1)		
Pretreatment QMG, Mean (standard deviation)	17.04 (3.83)	25
Pretreatment QMG, Median [25%, 75%]	17 [14; 20]	25
V1 MG-QOL, Mean (standard deviation)	25.93 (13.42)	43
V1 MG-QOL, Median [25%; 75%]	23 [15; 36]	43
V1 ADL, Mean (standard deviation)	5.84 (2.92)	45
V1 ADL, Median [25%; 75%]	6 [4; 7]	45
V1 QMG, Mean (standard deviation)	14.4 (3.63)	45
V1 QMG, Median [25%; 75%]	14 [11; 16]	45
V1 MG-Composite, Mean (standard deviation)	9.6 (5.44)	45
V1 MG-Composite, Median [25%; 75%]	8 [6; 11]	45
V1 Neuro-QoL fatigue, Mean (standard deviation)	44.44 (19.31)	43
V1 Neuro-QoL fatigue, Median [25%; 75%]	39 [28.5; 59]	43

Modified from Diez-Porras et al. 2020 [2]. Abbreviations: MGFA, Myasthenia Gravis Foundation of America; CT, computed tomography; QMG, Quantitative Myasthenia Gravis Scoring System; ADL, Myasthenia Gravis-Activities of Daily Living Profile; MG-Composite, Myasthenia Gravis-Composite Scale.

**Table 2 jcm-11-02189-t002:** Comparison of the mean MG-QOL15 score at the different visits, using a post hoc test based on the last model.

	Visit 1		4 Weeks		6 Weeks		4 Weeks–Visit 1			6 Weeks–Visit 1			6 Weeks–4 Weeks		
	Mean	CI	Mean	CI	Mean	CI	Dif	CI	Pval	Dif	CI	Pval	Dif	CI	Pval
MG-QOL15	25.93	[22.32, 29.54]	14.91	[11.3, 18.52]	10.53	[6.93, 14.14]	−11.02	[−15.05, −7]	<0.0001	−15.4	[−19.42, −11.37]	<0.0001	−4.37	[−8.4, −0.34]	0.0301

Abbreviations: CI, confidence interval; Dif, difference; pval, *p* value.

## Data Availability

All data are available under request to Carlos Casasnovas (carloscasasnovas@bellvitgehospital.cat).

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
