# Peer review of "Quality of Life in Myasthenia Gravis and Correlation of MG-QOL15 with Other Functional Scales"

_jcm, 2022, doi:10.3390/jcm11082189_

Round 1
Reviewer 1 Report
Porras et al present and nice review of their experience with the MG-QOL15 scale. The scale is a well established easy to administer scale with extensive use. the scale has longitudinal validity. In 2016 it was updated to the MG-QOL15r. This study further validates the scale in the tested subjects. The study is nicely done, but does not add to the body of knowledge on this scale.
- The study was developed in 2016 and at that time the QOL15 was the best scale. Currently the QOL15r is in use and not mentioned in the study.
- Given that the scale is validated, the authors need to more clearly define what new analysis they added. Has anyone studied IVIg and the scale? A better introduction and discussion of this is needed.
- 4 subjects were double negative for Anti-RAch. How were they diagnosed, SFEMG? Did separating them from the analysis impact results?
- There is no CONSORT chart or checklist, this needs to at least be in the supplemental data section.
Author Response
Please find attached the answers in the attached file.
Best regards,
Laura Diez Porras

Reviewer 2 Report
In general, this study described a study that using scales to evaluate the Health related quality of life (HRQOL) in myasthenia gravis (MG) after intensive treatment. The assessment scale is comprehensive, and the assessment process is well-trained. Also, this study differentiated the subtypes of MG, which showed a relative intact data. Moreover, the authors objectively discuss the limitations of this study, demonstrating a deep knowledge of this area. However, there are several suggestions for the improvement of this article. 1. It will be better to enumerate the content in Method and Result parts, which may benefit our reading and understanding. 2. The Table 1 and Table 2 should be rearranged, which is currently disordered.Author Response
Please find attached the answers in the attached file.
Best regards,
Laura Diez Porras
